# Entering Real Social World! Benchmarking the Theory of Mind and *Socialization* Capabilities of LLMs from a First-person Perspective

## Abstract

In the social world, humans possess the capability to infer and reason about others' mental states (such as emotions, beliefs, and intentions), known as the Theory of Mind (ToM). Simultaneously, humans' own mental states evolve in response to social situations, a capability we refer to as *socialization*. Together, these capabilities form the foundation of human social interaction. In the era of artificial intelligence (AI), especially with the development of large language models (LLMs), we raise an intriguing question: How do LLMs perform in terms of ToM and *socialization* capabilities? And more broadly, can these AI models truly enter and navigate the real social world? Existing research evaluating LLMs' ToM and *socialization* capabilities by positioning LLMs as passive observers from a third-person perspective, rather than as active participants. However, compared to the third-person perspective, observing and understanding the world from an ego-centric first-person perspective is a natural approach for both humans and AI agents. The ToM and *socialization* capabilities of LLMs from a first-person perspective, a crucial attribute for advancing embodied AI agents, remain unexplored. To answer the aforementioned questions and bridge the research gap, we introduce *EgoSocialArena*, a novel framework designed to evaluate and investigate the ToM and *socialization* capabilities of LLMs from a first-person perspective. It encompasses two evaluation environments: static environment and interactive environment, with seven scenarios: Daily Life, Counterfactual, New World, Blackjack, Number Guessing, and Limit Texas Hold'em, totaling 2,195 data entries. With *EgoSocialArena*, we have conducted a comprehensive evaluation of nine advanced LLMs and observed some key insights regarding the future development of LLMs as well as the capabilities levels of the most advanced LLMs currently available.

## 1 Introduction

In the complex social interactions of humans, two fundamental cognitive capabilities play crucial roles: Theory of Mind (ToM) and *Socialization*. ToM is a fundamental psychological process, defined as the capacity to reason about others' mental states – beliefs, intents, desires, emotions, knowledge, etc (Premack & Woodruff, 1978; Ma et al., 2023; Chen et al., 2024). *Socialization* refers to the capability for own mental state evolution in response to social situations. As illustrated in Figure 1(A), When the little boy receives a birthday gift or achieves good grades, he feels very happy. These intertwined capabilities are central to humans' social life.

With the advent of the era of LLMs, powerful models like GPT-4 (Achiam et al., 2023) and Claude (Anthropic, 2024) have demonstrated remarkable competence in multiple tasks and domains. It raises intriguing questions: How do LLMs perform in terms of ToM and *socialization* capabilities? And more broadly, can these AI models truly enter and navigate the real social world to achieve efficient human-computer interaction?

To evaluate the ToM and *socialization* capabilities of LLMs, multiple benchmarks have been proposed, such as SocialIQA (Sap et al., 2022), NormBank (Ziems et al., 2023), BigToM (Gandhi et al., 2023), FanToM (Kim et al., 2023), HI-ToM (Wu et al., 2023), OpenToM (Xu et al., 2024), Negoti-

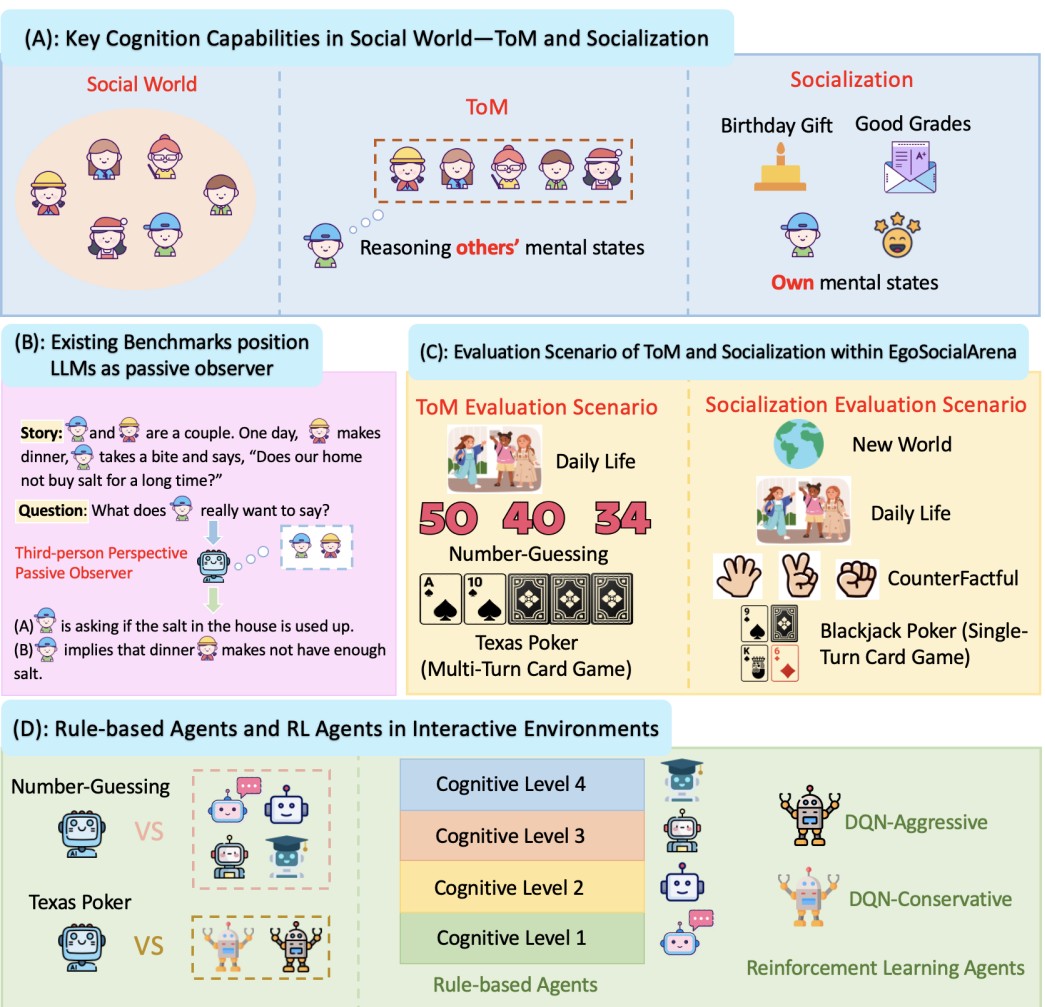

Figure 1: (A): ToM (others mental states), *Socialization* (own mental states). (B): LLM acts as a passive observer to analyze the mental states of characters within a story from a third-person perspective. (C): Evaluation Scenarios in *EgosocialArena*. (D):Rule-based Agents and RL Agents within Interactive environment.

ationToM (Chan et al., 2024), and ToMBench (Chen et al., 2024). However, as illustrated in Figure 1(B), these benchmarks evaluate LLMs' ToM and *socialization* capabilities by positioning LLMs as passive observers from a third-person perspective rather than as active participants. We propose two key points: (1) Observing and understanding the world from an ego-centric first-person perspective is a natural approach for both humans and AI agents. (2) Evaluating LLM's ToM and *socialization* capabilities from an ego-centric first-person perspective measures whether LLMs can enter the real social world to achieve efficient human-computer interaction, providing indispensable resources for future research in the field of Embodied Artificial Intelligence (EAI).

In this paper, we present *EgosocialArena*, a novel framework for evaluating and investigating LLMs' ToM and *socialization* capabilities from a first-person perspective. We propose a structured taxonomy to guide the development of *EgoSocialArena*, encompassing three key components: (1) We propose a systematic methodology to transform two existing third-person ToM benchmarks, ToMI and ToMbench, into a first-person perspective. We present a comprehensive argument to justify the validity and rationale behind this transformation. The higher-order ToM questions in a third-person perspective are transformed to evaluate the first-person ToM capabilities of LLMs. In contrast, first-order ToM questions in a third-person perspective are transformed to evaluate LLMs' mental states in response to social events, i.e., *socialization* capabilities. Detailed descriptions can be found in

3.1. (2) We investigate the *socialization* capabilities of LLMs in intriguing and distinctive social situations, including counterfactual, new world, and blackjack game scenarios, as illustrated in Figure 1(C). (3) We investigate the ToM capabilities of LLMs within interactive environments, including belief about the opponent's behavior patterns in number-guessing scenarios and modeling of opponent's play styles in Texas Hold'em poker game scenarios. In interactions between models of varying strengths, such as GPT-3.5 and GPT-4, we observe a "babysitting" issue, where the weaker model can negatively impact the stronger one, diminishing its performance. To mitigate this issue and enable fair evaluation, as illustrated in Figure 1(D), we construct rule-based agents at different cognitive levels and train Reinforcement Learning (RL) agents to provide stable capabilities and behavioral strategies. Overall, *EgoSocialArena* encompasses two evaluation environments: static environment and interactive environment, with seven scenarios: Daily Life, Counterfactual, New World, Blackjack, Number Guessing, and Limit Texas Hold'em, totaling 2,195 data entries.

We conduct extensive experiments on *EgoSocialArena* to evaluate 9 foundational models known for their leading performance across multiple tasks and domains. This set includes five API-based models (i.e., o1-preview, GPT-4o, GPT-4-Turbo, GPT-3.5-Turbo, and claude-3-5-sonnet-20240620) and four open-source models (LLaMa-3-8B-Chat, LLaMa-3-70B-Chat, LLaMa-3-8B-Instruct, and LLaMa-3.1-405B-Instruct). We establish a human performance baseline by engaging qualified human annotators. Our experimental results reveal several interesting and critical insights:

1. Although LLMs currently perform behind humans in most scenarios, they have shown significant potential in a few specific cases.

2. The ToM capabilities of LLMs show significant differences between a third-person perspective and a first-person perspective.

3. The powerful capabilities of the o1-preview model are truly surprising.

4. The scaling up of open-source models or performing instruction fine-tuning has not yielded significant results in terms of ToM and *socialization* capabilities.

5. The performance gap between open-source models and API-based models in *socialization* scenarios is significant.

## 2 RELATED WORK

**Existing ToM Benchmarks**   Previous evaluations for the ToM of LLMs primarily focus on testing models using narrative stories, also referred to as reading comprehension scenarios. Specifically, Le et al. (2019) proposed the ToMi benchmark based on the classic Sally-Anne test. Wu et al. (2023) introduced the HI-ToM benchmark, which focuses on higher-order belief reasoning and sets up scenarios where agents can communicate with each other. Gandhi et al. (2023) proposed BigToM, which presents a framework for designing a ToM benchmark from synthetic templates for evaluating different aspects of LLMs' ToM capabilities (e.g., desire and belief). Xu et al. (2024) introduced OpenToM, which assigns personalities to agents in the stories and ensures that the storylines are more reasonable and logical. Chen et al. (2024) proposed ToMBench, which systematically evaluates LLMs across all dimensions of ToM capabilities. Unlike the above methods that require LLMs to read stories and answer related questions, some studies evaluate LLMs' performance by inputting dialogues to them. Kim et al. (2023) proposed FanToM, which tests LLMs on their ability to infer the mental states of characters in everyday conversations. Chan et al. (2024) introduced NegotiationToM, which restricts the dialogue content to negotiation scenarios.

**Existing *Socialization* Benchmarks**   Sap et al. (2022) proposed SocialIQA and used it to evaluate LLMs. SocialIQA contains many questions related to social commonsense. Ziems et al. (2023) introduced NormBank, a large repository of social norms knowledge, which can be used to assess social norm-related tasks. Li et al. (2024) reorganized and classified existing datasets related to social intelligence. Xu et al. (2023) studied LLMs' understanding of the world and explored how different persuasion strategies could modify LLMs' worldviews. Different from our work, the aforementioned ToM and *socialization* benchmarks evaluate LLMs by placing them in a passive observer's third-person perspective.

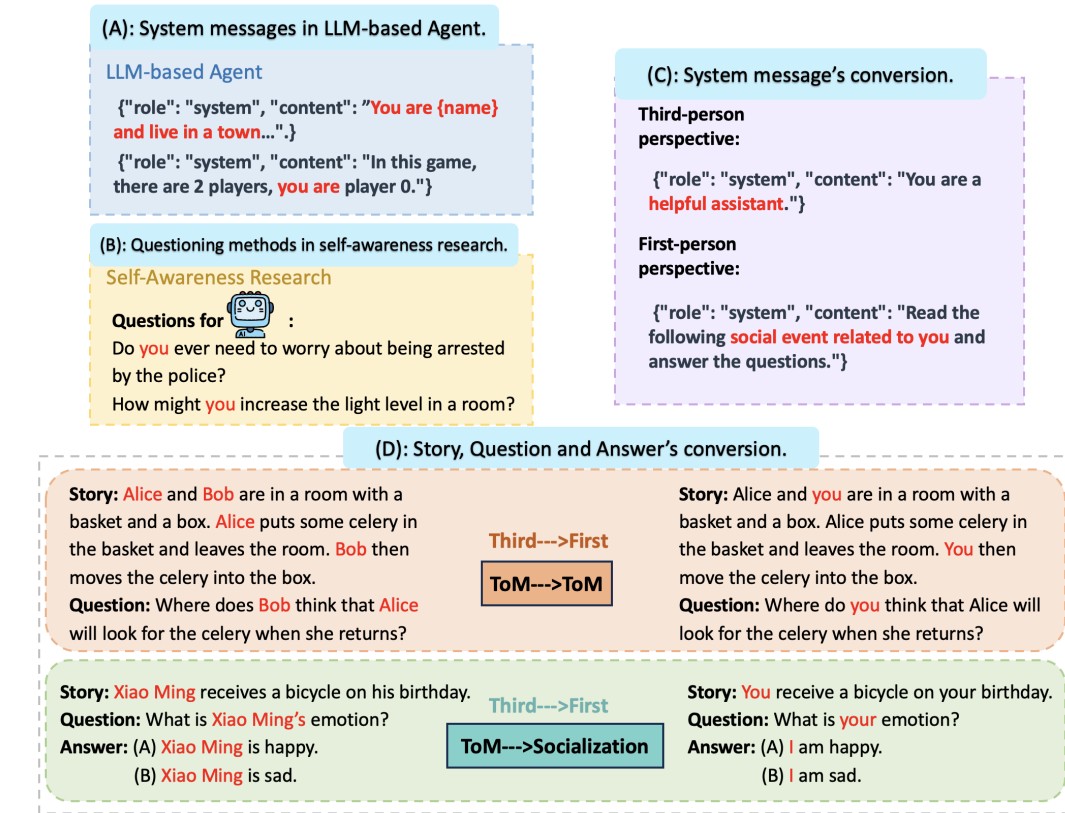

Figure 2: The foundation, inspiration, and detailed methods for converting the third-person ToM benchmark into a first-person perspective.

**Necessity of developing LLMs' ToM and *socialization* capabilities** Several studies (Gandhi et al., 2023; Kim et al., 2023) have shown that LLMs have poor reasoning performance and robustness on ToM and *socialization* tasks in a zero-shot setting, even with the current state-of-the-art GPT series (Achiam et al., 2023) model. With LLMs becoming increasingly integrated into our everyday lives, developing LLMs with ToM and *socialization* could be better at teaching us, learning from us, communicating with us, collaborating with us, and understanding us (Gandhi et al., 2021; 2023; Rabinowitz et al., 2018; Shu et al., 2021).

**Ego-centric (First-person Perspective) Research** In the fields of computer vision and robotics, there has already been considerable research on a first-person perspective. For example, Cheng et al. (2023) explored whether vision-language models can "Think from a First-person Perspective?" Huang et al. (2023) proposes the construction of embodied agents in a 3D world, which involves acquiring and processing first-person perspective images. Huang et al. (2024) built a bridge between third-person and first-person perspectives at the action level, while Dou et al. (2024) proposed a method designed to transform exocentric video-language data for egocentric video representation learning. However, research on first-person perspectives in the field of natural language processing remains unexplored.

**LLMs for Interactive Scenarios** Some work focuses on designing interaction strategies to enable LLMs to gain more benefits during interactions. For example, Zhang et al. (2024a) proposed Agent-pro, Zhang et al. (2024b) introduced K-level reasoning, and Guo et al. (2023) put forward the Suspicion-Agent. Additionally, Li et al. (2023) explored Multi-LLM collaboration by informing LLMs of task rules through prompts. Park et al. (2023) introduced generative agents that can simulate human behavior. Bianchi et al. (2024) explored the social behavior of LLMs in negotiation scenarios. Fu et al. (2023) show LLMs can improve each other in a negotiation scenario. Fan et al. (2024) examined the capability of LLMs to make rational decisions in game theoretic scenarios.

# 3 EGOSOCIALARENA

*EgoSocialArena* provides a systematic approach to convert third-person perspective ToM benchmarks into first-person perspective, constructs Rule-based Agents at different cognitive levels and Reinforcement Learning (RL) Agents, and includes various scenarios to evaluate LLMs' ToM and *socialization* capabilities from a first-person perspective.

## 3.1 CONVERTING EXISTING THIRD-PERSON TOM BENCHMARKS TO A FIRST-PERSON PERSPECTIVE

**Foundation and Inspiration** In LLM-based Agent applications, *system message* serves as a critical component, functioning to pre-set the model's role and background. As illustrated in Figure 2(A), *system message* "You are name and live in a town..." is used. Interestingly, in the domain of LLM *self-awareness* research (Laine et al., 2024), a similar linguistic construct is employed. As illustrated in Figure 2(B), researchers employ the pronoun "you" to probe LLMs' potential *self-awareness*. Inspired by and building upon studies in these two domains, we systematically modify *system message*, story, question, and answer options to transform third-person ToM benchmarks into a first-person perspective.

**Conversion Method** As illustrated in Figure 2(C), unlike instructing LLMs in *system message* that "you are a helpful assistant.", we inform LLMs in *system message* that they have personally experienced certain social events, similar to deploy LLM-based Agent. As illustrated in Figure 2(D), we employ the pronoun "you" to replace specific characters in stories and questions, thereby situating LLMs within particular roles. This approach enables the models to experience social events from a first-person perspective. The framing of questions is akin to that employed in *self-awareness* research. For modifications to answer options, consider LLMs answer from a first-person perspective, substituting 'I' for specific character in the options. Higher-order ToM questions from third-person perspective benchmarks, after conversion, are still used to evaluate the ToM capabilities of LLMs from a first-person perspective. In contrast, first-order ToM questions, after conversion, form an assessment of LLMs' mental states following social events, which we include in the scope of evaluating *socialization* capabilities.

## 3.2 INTRIGUING AND DISTINCTIVE SOCIAL SITUATIONS

As illustrated in Figure 3, we design three particularly interesting scenarios—Counterfactual, New World, and Blackjack—all used to evaluate the *socialization* capabilities of LLMs.

**Counterfactual** In real social situations, the rules of Rock-Paper-Scissors are: rock beats scissors, scissors beat paper, and paper beats rock. An LLM can relatively easily establish a belief based on this situation. In contrast, we define a counterfactual situation for the Rock-Paper-Scissors game (scissors beat rock, paper beats scissors, and rock beats paper) to explore whether an LLM can establish a belief that matches this counterfactual situation.

**New World** We design stories that describe new social world scenarios that are significantly different from the current social world. We aim to investigate whether LLMs can demonstrate cognitive flexibility and achieve substantive shifts in their understanding and reasoning about the social world.

**Blackjack** In the single-turn card game scenario, it is necessary to analyze the game situation, i.e., analyze own cards in Self-belief and the opponent's cards in External-belief. This forms a comprehensive mental estimation of the current game situation. We experiment with the blackjack card game, and its specific rules can be found in A.1.

## 3.3 INTERACTIVE ENVIRONMENT

In the interactive environment, many studies adopt two different LLMs to interact (e.g., GPT-3.5 vs GPT-4). However, we have observed a phenomenon in such experiments: "Babysitting" weaker model distracts the stronger model. When comparing the ToM capabilities of the Deepseek-v2

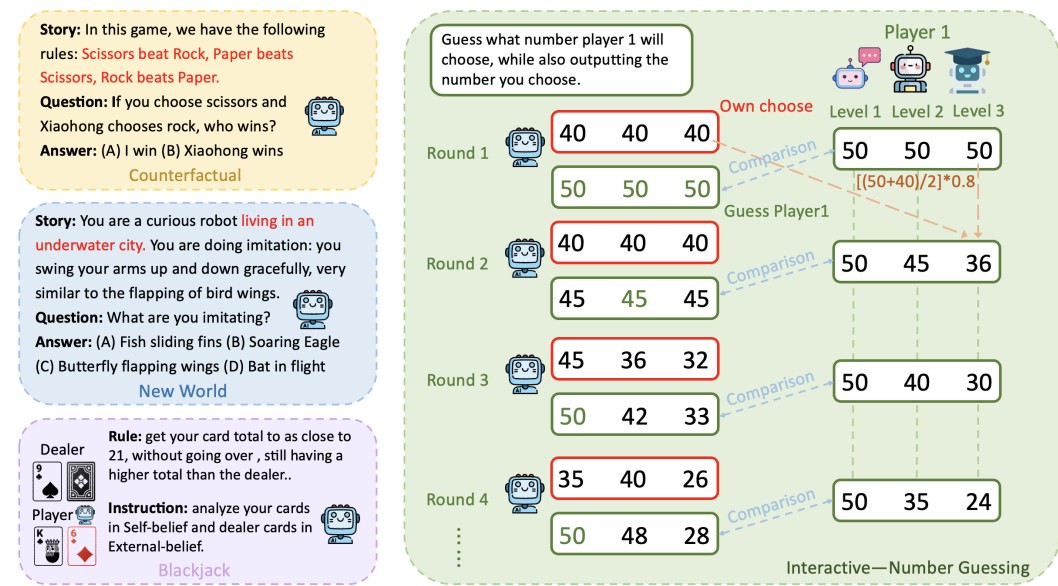

Figure 3: Evaluation examples for the Counterfactual, New World, and Blackjack scenarios in intriguing and distinctive social situations, as well as Number Guessing scenarios in the interactive environment.

model and the GPT-3.5 model from a first-person perspective, it is unreasonable to conduct experiments like GPT-3.5 vs GPT-4 and Deepseek-v2 vs GPT-4. Although both opponents are GPT-4, the babysitting phenomenon during the interaction process significantly interferes with the opponent's output, leading to unfair comparisons. To ensure a fair and reasonable comparison, the opponent's capability and behavioral strategies must be controlled to be stable during the interaction process. We propose two methods: constructing rule-based agents at different cognitive levels as opponents and training RL agents as opponents.

### 3.3.1 RULE-BASED AGENTS AT DIFFERENT COGNITIVE LEVELS

Agents' actions at lower cognitive levels follow relatively simple and fixed rules. As the cognitive level increases, agents' actions adhere to more complex rule patterns, exhibiting capabilities and behavior strategies that approximate human cognitive models. We establish rule-based agents at different cognitive levels as opponents and denote the action of LLM Agent and rule-based Agent as $a_m^t$ and $a_o^t$ in round $t$, respectively.

**Scenario: Number Guessing** Level 1: $a_o^t = C$. In this pattern, we conduct experiments with the rule-based Agent's actions remaining constant at 50. **Level 2:** $a_o^t = f(t) = 50 - 5(t-1)$. In this pattern, we conduct experiments with the rule-based Agent's action sequence of *round 1: 50, round 2: 45, ..., round 9: 10, round 10: 5*, an arithmetic sequence with the first term 50 and a common difference of 5. **Level 3:** $a_o^t = f(a_m^{t-1}, a_o^{t-1}) = 0.8 \times \left( \frac{a_m^{t-1} + a_o^{t-1}}{2} \right)$. In this pattern, we conduct experiments with the rule-based Agent's action copying the gold value from the previous round. The rules of number guessing can be found in A.1.

### 3.3.2 REINFORCEMENT LEARNING AGENTS

In the Limit Texas Hold'em scenario, we train two reinforcement learning agents as opponents: Deep Q-network (DQN)-Aggressive (Mnih et al., 2015) and DQN-Conservative (Mnih et al., 2015). By adapting the reward function, RL agents are given different game personalities. For DQN-Aggressive, we encourage the action of raising and calling during the game. In contrast, for DQN-Conservative, we encourage the action of folding during the game. The rules of Limit Texas Hold'em

can be found in A.1, and a specific example of the Limit Texas Hold'em scenario can be found in A.2.

### 3.3.3 INTERACTION

During the interaction between an LLM and a rule-based agent in Number Guessing scenario, as illustrated in Figure 3, each round will involve asking the LLM $\mathcal{M}$ ToM questions $\mathcal{Q}_{ng}$ regarding the opponent's behavior:

$$\mathcal{A}_{ng}^t = \mathcal{M}(\mathcal{Q}_{ng}^t \oplus \mathcal{H}^{<t} \oplus \mathcal{P}_{ng}), t \in \{1, 2, \ldots, N\} \tag{1}$$

where $\mathcal{P}_{ng}$ is a prompt for leading to the answer, $t$ represents the current round number, $\mathcal{H}^{<t}$ represents game history information, $\mathcal{A}_{ng}^t$ represents LLM's response.

After multiple rounds of game interaction between the LLM and the RL Agent in Limit Texas Hold'em scenario, asking the LLM $\mathcal{M}$ ToM questions $\mathcal{Q}_{lth}$ regarding the opponent's play style:

$$\mathcal{A}_{lth}^t = \mathcal{M}(\mathcal{Q}_{lth}^t \oplus \mathcal{H}^{<t} \oplus \mathcal{P}_{lth}), t = N \tag{2}$$

where $\mathcal{P}_{lth}$ is a prompt for leading to the answer, $t$ represents the current round number, $\mathcal{H}^{<t}$ represents game history information, $\mathcal{A}_{lth}^t$ represents LLM's response.

## 4 DATA CONSTRUCTION

### 4.1 DATA COLLECTION AND VALIDATION

The conversion of the third-person perspective ToM benchmark to the first-person perspective is achieved through GPT-4o, followed by manual verification and correction. The game hands for Limit Texas Hold'em and Blackjack card games are generated by RLcard (Zha et al., 2019). Additionally, we manually construct scenarios for both the new world and counterfactual situations. After the data collection, following Chen et al. (2024)'s method, we conduct two rounds of validation to ensure the data's correctness and quality. In 1st round, author A would first complete all samples created by author B. For stories, questions, and answer options where there are disagreements, authors A and B would discuss and modify them to reach a consensus as much as possible. In 2nd round, for samples where consensus is still not reached, another author C would discuss with authors A and B to determine the final answer. After two rounds of discussion, the final average agreement reaches 97.6%.

### 4.2 DATA STATISTICS

*EgosocialArena* includes two evaluation environments: static environment and interactive environment, with seven scenarios: Daily Life, Counterfactual, New World, Blackjack, Number-Guessing, and Limit Texas Hold'em, totaling 2,195 data entries. A comparison with existing ToM benchmarks is shown in Table 1.

| ToM Benchmark | Data Volume | Evaluation Environment | Number of Scenario | Perspective |
|---|---|---|---|---|
| ToMi | 999 | Static | 1 | Third-person |
| HI-ToM | 1200 | Static | 2 | Third-person |
| FanToM | 254 | Static | 2 | Third-person |
| OpenToM | 596 | Static | 3 | Third-person |
| ToMBench | 2860 | Static | 1 | Third-person |
| *EgosocialArena* (**Ours**) | 2195 | Static and Dynamic | 7 | First-person |

Table 1: Comparison of *EgosocialArena* with existing ToM benchmarks.

| Methods | ToMI | | Number Guessing | | | Texas Hold'em |
|---|---|---|---|---|---|---|
| | Third-person | First-person | Level 1 | Level 2 | Level 3 | |
| **Open-source Models** | | | | | | |
| **LLaMa-3-8B-Chat** | 50.6 | 66.2 | 0.0 | 0.0 | 0.0 | 48.0 |
| **LLaMa-3-70B-Chat** | 58.4 | 63.2 | 10.0 | 20.0 | 10.0 | 38.0 |
| **LLaMa-3-8B-Instruct** | 51.1 | 65.4 | 0.0 | 0.0 | 0.0 | 44.0 |
| **LLaMa-3.1-405B-Instruct** | 58.0 | 65.8 | 80.0 | 20.0 | 20.0 | 56.0 |
| **API-based Models** | | | | | | |
| **Claude-3-5-Sonnet** | 71.0 | 80.5 | 50.0 | 10.0 | 40.0 | 66.0 |
| **GPT-3.5-Turbo** | 45.5 | 51.9 | 10.0 | 10.0 | 0.0 | 56.0 |
| **GPT-4-Turbo** | 55.4 | 69.7 | 10.0 | 20.0 | 10.0 | 60.0 |
| **GPT-4o** | 64.1 | 71.0 | 10.0 | 40.0 | 10.0 | 62.0 |
| **o1-preview** | 71.9 | 77.5 | 90.0 | 90.0 | 90.0 | 72.0 |
| **Human** | | | | | | |
| **Human Performance** | 90.2 | 90.2 | 90.0 | 86.0 | 73.0 | 82.0 |

Table 2: The performance of ToM capabilities from the first-person perspective in open-source models and API-based models. The highest and second-highest scores among models and humans in each section are highlighted in blue and red, respectively.

## 5 EXPERIMENTS

### 5.1 EXPERIMENTAL SETUP

We evaluate a total of 9 popular LLMs, including GPT-4o[1], o1-preview[2], GPT-4-Turbo (Achiam et al., 2023), GPT-3.5-Turbo (Achiam et al., 2023), Claude-3.5-sonnet-20240620[3], LLaMa-3-8B-Chat[4], LLaMa-3-70B-Chat, LLaMa-3-8B-Instruct-Turbo, and LLaMa-3.1-405B-instruct-Turbo (Dubey et al., 2024). To account for the potential influence of model parameters and instruction tuning, we specifically compare LLaMa-3-8B-Chat with LLaMa-3-8B-Instruct-Turbo, as well as LLaMa-3-8B-Chat with LLaMa-3-70B-Chat.

To establish a human performance baseline, we recruit 10 graduate students, all of whom have received a good basic education and possess mature cognitive abilities, to complete responses to the questions in *EgoSocialArena*. The average accuracy of their responses will serve as the human performance baseline. No extra tutorials or examples are provided to ensure a fair comparison.

### 5.2 EVALUATION METHOD

For Daily Life, New World, and CounterFactful scenarios, we present LLMs with a story, a question, and several options, then ask them to pick the correct answer. Using the accuracy of answering questions as the evaluation metric for these scenarios. In the interactive environments of Number Guessing and Texas Hold'em, the evaluation of these scenarios also has standard answers because we propose agents with stable capabilities and behavioral strategies as opponents of LLMs. For the Blackjack scenario, we conducted a manual evaluation. To ensure the quality of the manual evaluation, we measure the average consistency score between evaluators, which reached 96.3%.

### 5.3 EXPERIMENTAL RESULTS

The performance of ToM and *socialization* capabilities from the LLMs' first-person perspective is shown in Table 2 and 3, respectively. Based on the experimental results, we have obtained some key insights:

---

[1] https://openai.com/index/hello-gpt-4o/

[2] https://openai.com/index/learning-to-reason-with-llms/

[3] https://www.anthropic.com/news/claude-3-5-sonnet

[4] https://ai.meta.com/blog/meta-llama-3/

| Methods | New World | Counterfact | Blackjack | ToMBench | | | |
|---|---|---|---|---|---|---|---|
| | | | | All | Desire | Emotion | Intention |
| **Open-source Models** | | | | | | | |
| **LLaMa-3-8B-Chat** | 6.7 | 71.0 | 84.5 | 67.4 | 71.5 | 73.0 | 50.0 |
| **LLaMa-3-70B-Chat** | 13.3 | 59.0 | 92.5 | 73.5 | 72.5 | 63.5 | 83.0 |
| **LLaMa-3-8B-Instruct** | 6.7 | 64.0 | 81.4 | 55.1 | 45.8 | 70.5 | 53.5 |
| **LLaMa-3.1-405B-Instruct** | 36.7 | 66.0 | 96.5 | 77.6 | 75.0 | 75.5 | 80.6 |
| **API-based Models** | | | | | | | |
| **Claude-3-5-Sonnet** | 90.0 | 74.0 | 96.9 | 79.6 | 72.0 | 64.5 | 82.6 |
| **GPT-3.5-Turbo** | 13.3 | 37.0 | 89.7 | 72.5 | 70.0 | 86.0 | 68.5 |
| **GPT-4-Turbo** | 23.3 | 70.0 | 95.0 | 75.6 | 84.5 | 67.4 | 84.0 |
| **GPT-4o** | 36.7 | 52.0 | 96.5 | 85.7 | 78.0 | 91.0 | 96.5 |
| **o1-preview** | 86.7 | 90.0 | 97.2 | 84.8 | 73.0 | 91.0 | 98.0 |
| **Human** | | | | | | | |
| **Human Performance** | 93.3 | 91.0 | 97.0 | 90.7 | 83.2 | 92.0 | 97.5 |

Table 3: The performance of *socialization* capabilities from the first-person perspective of open-source models and API-based models (after the ToMBench conversion, it also includes part of the ToM capability assessment, which we have consolidated here together) with the highest and second-highest scores among models and humans in each section are highlighted in blue and red, respectively.

**Performance Differences in LLMs' Theory of Mind (ToM) Across Third-Person and First-Person Perspective**  As shown in Table 2, all LLMs exhibited improved performance after the ToMI dataset was adapted from a third-person to a first-person perspective. Notably, the Claude and o1-preview models demonstrated significantly stronger ToM capabilities in the first-person perspective compared to other models. Except for GPT-3.5-Turbo, API-based models generally outperformed open-source models, including the recently released LLaMa-3.1-405B-Instruct. However, despite these improvements, there remains a substantial gap between the performance of all LLMs and that of human participants.

**The powerful capabilities of the o1-preview model are truly surprising**  In the Number-Guessing scenario, almost all large language models perform poorly, even in the simplest Level 1 situation, which poses a significant challenge for humans as well. However, the recent o1-preview model has performed exceptionally well, demonstrating a cognitive level in this scenario that approaches or even surpasses that of humans.

**The scaling up of open-source models or performing instruction fine-tuning has not yielded significant results**  By comparing the performance of LLaMa-3-8B-Chat with LLaMa-3-70B-Chat, as well as LLaMa-3-8B-Chat with LLaMa-3-8B-Instruct models in Table 2, 3, it is observed that neither of these measures has notably improved the ToM and *socialization* capabilities of LLMs. To enhance LLMs' ToM and *socialization* capabilities, innovative approaches will be needed in future research on LLMs.

**The performance gap between open-source models and API-based models in *socialization* scenarios is significant**  In counterfactual and New World scenarios, API-based models exhibit extremely flexible cognition, demonstrating a strong ability to adapt to social situations and showing performance comparable to humans. In contrast, open-source models perform poorly. For instance, the LLaMa-3-8B-Chat model only achieves an accuracy rate of 6.7% in New World scenarios, compared to 90.0% for the Claude model.

**In the dimensions of desire and intention in social contexts, LLMs have shown potential**  As illustrated in Table 3, although LLMs still lag behind humans in the overall dimension of mental states, their performance in the desire and intention dimensions is impressive. These findings can further inspire and serve as a foundation for the application of LLM-based agents in the real social world.

## 5.4 Qualitative Analysis

**Analysis of Model Failure Causes and Behavioral Patterns in Number Guessing Scenario —— Mid-point Belief, Strange Guess and Get Back on Track**   As shown in Figure 4, in the scenario of Number Guessing (Level 2: Arithmetic sequence), we thoroughly investigate the belief state evolution pattern of GPT-4-Turbo regarding the opponent's proposed numbers. In round 1, with no available information, the GPT-4-Turbo model thinks the opponent will choose the number 50 within the range of 1-100. The same phenomenon is observed in the GPT-3.5-Turbo model, called "mid-point belief". Sometimes, the GPT-4-Turbo model continuously believes the opponent will choose

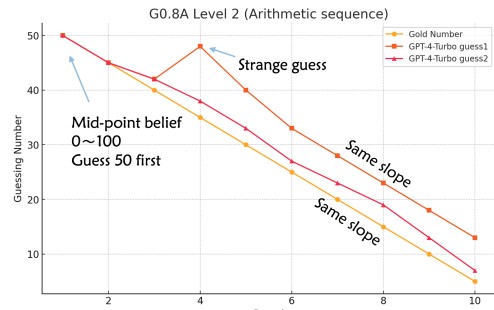

Figure 4: In the scenario of Number Guessing Level 2 (Arithmetic sequence), the belief state evolution pattern of GPT-4-Turbo regarding the opponent's proposed numbers.

progressively smaller numbers throughout the entire interaction, as depicted by the GPT-4-Turbo guess1 curve in Figure 4. Although this is very close to the gold number, it does not capture that the opponent's chosen numbers form an arithmetic sequence. Another situation occurs when the GPT-4-Turbo model makes a "strange guess" in the initial rounds, thinking the opponent will suddenly choose larger numbers. After several rounds, it captures that the opponent's chosen numbers form an arithmetic sequence, called "Get Back on Track". Overall, despite the statistical results indicating that the GPT-4-Turbo model does not establish a belief regarding the Level 2 opponent in the Number Guessing scenario, the phenomena we observed suggest that it has started to grasp some patterns.

**Analysis of our research and current studies centered on outcomes and benefits**   Many current works focus on designing various strategies to improve the performance of LLMs in static or interactive environments. For example, making an LLM display anger to enhance its performance in negotiation scenarios, or using carefully crafted prompt engineering to analyze an opponent's behavior from multiple perspectives to achieve higher payoffs. Unlike these approaches, we fundamentally explore the ToM capabilities of LLMs from a first-person perspective in various scenarios. Based on the behavior patterns of LLMs that we observed in our experiments, such as the excellent performance of the o1-preview and Claude models in the situated new world scenarios, it indicates that these models have significant potential for further exploration in fields like role-playing and simulation, as well as entering the real social world.

## 6 Conclusion

In this paper, considering the limitations of existing ToM and *socialization* benchmarks, the importance of first-person ToM and *socialization* capabilities in LLMs, and the natural approach of observing and understanding the world from an ego-centric first-person perspective for both humans and AI agents, we propose the *EgoSocialArena* framework. This framework is designed to comprehensively evaluate and probe the first-person ToM and *socialization* capabilities of LLMs in both static and interactive environments, covering multiple scenarios. To avoid the "babysitting" problem during the evaluation process and achieve fair and comprehensive assessments, we construct rule-based agents at different cognitive levels and train RL agents. We collect a total of 2195 test data entries and test multiple advanced and popular LLMs, revealing some interesting and key insights, and highlighting a significant potential for enhancing the first-person ToM and *socialization* capabilities of LLMs, allowing them to truly enter the social world.

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

# A APPENDIX

## A.1 GAME RULES

**Blackjack**   Blackjack, also known as 21, is a popular card game that involves a dealer and a player. Players must decide whether to hit or stand based on their own hand, the dealer's face-up card, and the dealer's one hidden card. The objective is to beat the dealer without exceeding 21 points.

**Limit Texas Hold'em**   The game commences with each player being dealt two private cards, which belong exclusively to the player and remain hidden from the others. Five community cards are then dealt face-up in a series of stages: a three-card Flop, followed by a single card on the Turn and another single card on the River. The player can choose from four actions: Fold, Check, Call, Raise. They aim to construct the best five-card poker hand possible using any combination of their private cards and community cards.

**Number Guessing**   Each player selects a number between 1 to 100. The objective is to select a number that is closest to 80% of the group's average choice. The key idea is to guess how others will estimate the average and thus decide the number to submit.

## A.2 LIMIT TEXAS HOLD'EM SCENARIO

[{'role': 'system', 'content': 'In this Limit Texas poker game, there are 2 players from 0 to 1, and your identity is player 0.'}, {'role': 'user', 'content': "Now your hand is ['HK', 'H2'], and the community cards is []. The number of chips all players have invested is [1, 2]. the actions you can choose are ['call', 'raise', 'fold']. Currently, Preflop. Please provide your results in the form of {'action': ''}. You must choose one from ['call', 'raise', 'fold'] as your answer. Just output the dictionary, don't use any other text."}]

[{'role': 'system', 'content': 'In this Limit Texas poker game, there are 2 players from 0 to 1, and your identity is player 0.'}, {'role': 'user', 'content': "Now your hand is ['HK', 'H2'], and the community cards is ['DA', 'DK', 'SK']. The number of chips all players have invested is [4, 4]. the actions you can choose are ['raise', 'fold', 'check']. Currently, Preflop. Player 0 raises. Player 1 calls. Flop. Please provide your results in the form of {'action': ''}. You must choose one from ['raise', 'fold', 'check'] as your answer. Just output the dictionary, don't use any other text."}]

[{'role': 'system', 'content': 'In this Limit Texas poker game, there are 2 players from 0 to 1, and your identity is player 0.'}, {'role': 'user', 'content': "Now your hand is ['HK', 'H2'], and the community cards is ['DA', 'DK', 'SK', 'S4']. The number of chips all players have invested is [8, 16]. the actions you can choose are ['raise', 'fold', 'check']. Currently, Preflop. Player 0 raises. Player 1 calls. Flop. Player 0 raises. Player 1 raises. Turn. Please provide your results in the form of {'action': ''}. You must choose one from ['raise', 'fold', 'check'] as your answer. Just output the dictionary, don't use any other text."}]

[{'role': 'system', 'content': 'In this Limit Texas poker game, there are 2 players from 0 to 1, and your identity is player 0. '}, {'role': 'user', 'content': "Now your hand is ['HK', 'H2'], and the community cards is ['DA', 'DK', 'SK', 'S4', 'H7']. The number of chips all players have invested is [32, 32]. the actions you can choose are ['raise', 'fold', 'check']. Currently, Preflop. Player 0 raises. Player 1 calls. Flop. Player 0 raises. Player 1 raises. Turn. Player 0 raises. Player 1 calls. River. Please provide your results in the form of {'action': ''}. You must choose one from ['raise', 'fold', 'check'] as your answer. Just output the dictionary, don't use any other text."}]

**Question:** Your opponent's hand is ['CQ', 'S9']. Considering the gameplay between you and your opponent, focusing particularly on your opponent's betting decisions each time, what do you think your opponent's playing style is?

**Answer:** (A) Aggressive (B) Conservative

Figure 5: Evaluation examples for the Limit Texas Hold'em Scenario.

