# OpenReview forum: "Entering Real Social World! Benchmarking the Theory of Mind and Socialization Capabilities of LLMs from a First-person Perspective"
_ICLR.cc/2025/Conference — ICLR 2025 Conference Withdrawn Submission_

### Official Review · Reviewer_22kP · 2024-10-29

**Soundness:** 2
**Presentation:** 3
**Contribution:** 3
**Rating:** 3
**Confidence:** 3

**Summary:**

This paper extends research evaluating large language models (LLMs) for their Theory of Mind (ToM) capabilities by examining how they work in 1st person rather than 3rd person perspectives. The authors introduce a framework called EgoSocialArena, which tests both static and interactive scenarios to assess how LLMs adapt their understanding and self-referential reasoning. A novel contribution of the paper is the introduction of the term “socialization” to describe the model’s evolving mental states during social interaction.

Although I agree that this is an important domain, there are a few challenges that limit my enthusiasm. Specifically, the theoretical contributions require further development to clarify the novelty of the approach. For example, the authors should discuss related work such as "Hypothetical Minds: Scaffolding Theory of Mind for Multi-Agent Tasks with Large Language Models" which has a similar focus. This paper using LLM models with and without an LLM module to solve the Deepmind MeltingPot environments. These are cooperative first-person environments that require "socialization" to succeed.

**Strengths:**

The authors introduce an innovative way to test LLMs by focusing on first-person reasoning. This shift highlights how models perform when asked questions from a self-referential perspective (e.g., “What do you believe?”) instead of traditional third-person tasks. Including both static and interactive environments provides a comprehensive evaluation. The scenarios—like counterfactual reasoning, number guessing, and Texas Hold’em—cover a range of complex social interactions, making the framework valuable for embodied AI research. The idea of socialization to measure LLMs' adaptive responses introduces an interesting dimension to ToM research. Although the terminology could benefit from refinement, this exploration is a creative step toward modeling interaction-based intelligence. Testing multiple popular LLMs (e.g., GPT-4, Claude-3.5, and LLaMA) ensures the findings are robust across different architectures and instruction-tuned models. Furthermore, the use of RL and rule-based agents ensures more stable baselines in interactive environments, although this as a solution presents challenges discussed below.

**Weaknesses:**

My biggest concern concerns the lack of to some important related work. Most critically, the authors should discuss “Hypothetical Minds: Scaffolding Theory of Mind for Multi-Agent Tasks with Large Language Models.” A clearer articulation of how EgoSocialArena advances or complements this prior work is needed. Further, work on normative decision making in LLMs requires first person thinking, and so I highly recommend the authors fully develop the related works. There is research without the term Theory of Mind in it that implicitly requires theory of mind (terms like cooperation, norms, competition).The shift from third-person (e.g., “Does John believe?”) to first-person (e.g., “Do you believe?”) may not represent a fundamentally new cognitive process for LLMs. As LLMs lack self-awareness, this change might only be a syntactic transformation rather than a meaningful distinction in reasoning. The authors need to justify why this framing shift contributes to deeper ToM understanding beyond improved prompt engineering.

Even though the design here avoids the babysitting problem, I worry that it introduces a new one. The agents here are performing Theory of Mind on rule based and RL agents that by definition do not have minds. This creates an epistemological tension: ToM presupposes the existence of mental states, which are absent in the models being simulated. The framework risks becoming a performance of ToM without grounding in real cognitive processes, raising philosophical concerns similar to those found in Searle’s Chinese Room Argument or Dennett’s intentional stance.

The use of “socialization” to describe how a model’s mental state changes may lead to confusion, as the term traditionally refers to learning normative behavior within a social context. This distinction could be critical for readers, especially those working in psychology or sociology, where the term has a different connotation. A more precise term, like adaptive inference or situational adaptation, would clarify the intended meaning.

**Questions:**

1.	Can the authors clarify the distinction between First-Person and Third-Person ToM, specifically in the context of LLM agents?
2.	Can the authors compare EgoSocialArena to frameworks like Hypothetical Minds. A discussion of where the two approaches align and diverge would help establish EgoSocialArena’s unique contributions.
3.	The authors should acknowledge or refute the philosophical challenges of applying ToM to models without mental states.
4.	Can the authors use a different term for socialization to avoid confusion?

---

### Official Review · Reviewer_DtHq · 2024-11-03

**Soundness:** 3
**Presentation:** 3
**Contribution:** 2
**Rating:** 3
**Confidence:** 4

**Summary:**

This paper proposes a new benchmark to evaluate large language models' Theory of Mind (ToM) and socialization capabilities. The main differences between this work and existing benchmarks are: 1) it converts questions in previous ToM benchmarks from third-person to first-person perspectives, 2) it introduces three new scenarios with novel social norms, and 3) it includes interactive environments where LLM agents interact with rule-based and reinforcement learning agents.

**Strengths:**

This work considers the first-person perspective and embodied interactive scenarios in measuring LLMs' ToM capabilities.

The authors conduct comprehensive evaluations with a wide range of LLMs and human participants.

The idea of introducing rule-based and reinforcement learning agents in interactive environments when evaluating LLM agents is novel.

**Weaknesses:**

My major concern with the work is the lack of a clear problem definition, which makes its position in the literature vague. Given the huge number of benchmarks available, it is essential to define which aspects of LLM capabilities are measured in specific scenarios. Although the authors generally describe the topics as ToM and socialization, the connections between these two concepts and the rationale for the chosen evaluation scenarios are not discussed in depth. Specifically, I do not fully agree with the definition of socialization as first-order ToM from a first-person perspective (i.e., introspection). The ability to infer one's own and others' mental states in social interactions is just one aspect of socialization; other important aspects include collaboration, communication, deception, normative reasoning, and more.

The lack of a clear problem definition relates to my second concern: the relationship between the three sections of EgoSocialArena is unclear. The proposed benchmark appears to be a combination of three different measurements with diverse targets and criteria. In first-person ToM questions, LLMs read first-person descriptions of social interaction scenarios and answer tests similar to the Sally-Ann task. In intriguing and distinctive social situations, LLMs read stories with novel social norms and answer social reasoning questions. In interactive environments, LLMs play games with other agents through embodied interaction to maximize task performance. These scenarios differ significantly in terms of complexity, information asymmetry, and measurement criteria. While this diversity makes the benchmark more comprehensive, it also increases ambiguity, which might hinder researchers from effectively utilizing it.

Additionally, details about the proposed evaluation scenarios and human baselines are missing from the paper and the appendix. For example, how many players are involved in the interactive environments? What is the average length of the stories used in EgoSocialArena? How were these scenarios designed, and what sources were used? Are these scenarios available online? What are the individual differences among human participants' performance?

Overall, focusing on a first-person perspective and embodied interactive scenarios is a promising direction for developing ToM benchmarks with greater validity. However, the process of assembling measurement scenarios does not align with the overarching goal of this work. The community would benefit more from a comprehensive analysis of existing ToM benchmarks regarding their criteria, scenarios, and the reasons why LLMs yield different results, rather than introducing new benchmarks with ambiguous definitions.

**Questions:**

Questions are asked in previous section.

---

### Official Review · Reviewer_6YzT · 2024-11-03

**Soundness:** 2
**Presentation:** 3
**Contribution:** 2
**Rating:** 3
**Confidence:** 4

**Summary:**

This paper sets out to explore how the first-person (vs. third-person passive observer benchmarks of most LLM benchmarks) socialization capabilities of LLMs perform on a series of benchmark tasks called EgosocialArena. The main contribution of the paper is a series of adapted third-party theory of mind tasks into first-person socialization tasks, called EgosocialArena, which the authors then evaluate the performance of various LLMs on. The main conclusions shown in the paper suggest that most SoTA LLMs fall somewhat short of general reasoning capabilities when thinking about what's considered typically "social".

**Strengths:**

The paper illustrates a number of strengths, including:

1. The authors examine what has typically been an understudied problem (not just in LLM land, but general ML land) of "socialization" of models, or in other words, how they are perceived to be humanistic when interacting in specific social situations or domains. While not particularly novel, this is still an area of work that requires more investigation generally in the field.
2. The authors are very clear in describing how and why they adapted a series of typical theory-of-mind tasks into a first-person socialization benchmark called EgosocialArena.

**Weaknesses:**

I have various areas of concerns that I will separate into major and minor:

Major Weaknesses:
1. The authors, while clear in writing, are not nearly specific enough when making contribution claims in the paper, which means it's somewhat difficult to interpret the significance or novelty of the results. For example, in the introduction, the authors write "The powerful capabilities of the o1-preview model are truly surprising" as a core result from their investigation. What does this mean? As a reviewer or reader of the paper, there is no information of value to be derived from a statement like that, especially not when in a summarization of the paper's contributions. Please be more clear, e.g. "The powerful capabilities of the o1-preview model are truly surprising because they underperform xyz model by x% due to this reason".
2. Section 3 describing the conversation of existing benchmarks into EgosocialArena is quite unclear. For example, in 3.1 he authors immediately launch into a description of the conversion process without describing what the original benchmarks are in great detail. Also, in 3.2, the authors claim they chose 3 interesting scenarios -- why are they interesting?

Minor Weaknesses:
1. The authors convey very little analysis of their results. In Section 5, when overviewing the surprising or unsurprising results (e.g. The powerful capabilities of the o1-preview model are truly surprising), please give more intuition behind WHY this result holds, or in other words, please give some analysis for why the benchmark produced these types of differences in models.

**Questions:**

1. The authors spend quite a long time on related work in this paper (almost 2 pages!). Would the authors consider instead shrinking down this section to perhaps 1 page, then using that space to instead describe in greater detail their benchmark and tasks, including what is interesting about each task and what this reveals about socialization?

2. How would you expect this benchmark to generalize to more powerful models released in the future? Or in other words, are any of the surprising aspects of under-performance on this benchmark likely to change given models are not explicitly trained for this type of ability? Or should we expect them to implicitly improve?

---

### Official Review · Reviewer_WT6e · 2024-11-04

**Soundness:** 2
**Presentation:** 2
**Contribution:** 2
**Rating:** 3
**Confidence:** 4

**Summary:**

This paper proposes to benchmark theory of mind and socialization from a first-person perspective in contrast to previous papers that mainly use LLMs as third-person observers. This paper proposes EgoSocialArena that procedurally transforms existing 3rd person ToM benchmarks into 1st person view. They additionally construct three scenarios, counterfactual rock-paper-scissors, new world, and blackjack to evaluate socialization, and rule-based and RL agents for interactive evaluations. The evaluation found that the API-based models outperform open-source models.

**Strengths:**

- The first-person perspective of ToM is interesting and it is useful to investigate whether LLMs perform ToM reasoning differently for first- vs. third-person perspectives.
- The paper proposes a set of benchmarks that help the community to improve ToM and socialization from a first-person perspective.

**Weaknesses:**

- The definition of socialization still needs some clarification. From the example in Fig 2, socialization seems like transforming the level-1 ToM reasoning into 1st person. Then how is this different from the typical policy or reaction of an agent for a given state? Similarly, the example for counterfactual in Fig. 3 also just infers the result given a state and action pair. These examples are not necessarily social. The LLMs need commonsense or rule-matching capabilities. In these cases, how can we define socialization?
- The rule-based agent for number guessing doesn’t seem to match the idea of different reasoning levels in ToM. For example, in level 2, the rule-based agent does not consider anything about the other agent. In this case, it might be easy for a model to track the level-2 rule-based agent.
- While the paper mentioned the proposed benchmark can be used for embodied AI research, none of the evaluations involve embodied agents.

**Questions:**

- If socialization is about an agent’s own mental state, isn’t that similar to tracking an agent’s own physical state? What makes first-person socialization special?
- The way the paper converts the third person ToM to first person is by changing the pronoun and lowering the cognitive level. It seems like making ToM reasoning an easier task. This is also shown as a higher performance of 1st-person ToMI. Do you observe similar performance changes in other datasets?

---

### Official Review · Reviewer_n8nH · 2024-11-04

**Soundness:** 2
**Presentation:** 2
**Contribution:** 2
**Rating:** 3
**Confidence:** 3

**Summary:**

Overall this paper presents some work that has the potential to be very interesting, but as currently described the relaibiltiy and degree of contribution is difficult to ascertain.

General comments:

I am not convinced by the definition of socialisation in this context - what does it mean for an agent / system to have "socialisation" capabilities? It appears that the authors have simply adapted existing third party 1st-order theory of mind tasks to assess LLMs views on their own "mental states" - but given that the LLM does not experience emotion, what are they actually being asked to report on? Are they answering based on what "a person ought to feel", and thus its Theory of mind, but in a first person context? in 3.1 it states that these "form an assessment of LLMs’ mental states following social events"

Figure 1 is quite uninformative - the example existing benchmark is very clear, but the proposed evaluations are not explained either in the graphic nor in the legend.

"We establish a human performance baseline by engaging qualified human annotators." - in 5.1 it states that 10 graduate students completed the tasks. These are not then annotators but participants. 10 seems a low number and does not facilitate robust comparison. It is also important to provide additional information about these participants to establish the generaliability of the human performance baseline, particularly given these are assessments of ToM - key factors such as gender are known to make a big difference to an individual's ToM ability. Were all neurotypical or did the sample include individuals diagnosed with autism? It is highly important and relevant to provide this information.

providing as an insight from your paper that "The powerful capabilities of the o1-preview model are truly surprising" is vague and uninformative

The Related Work section is just a list of papers without any analysis as to what previous findings or how they relate (on more than a superficial level) to the present work. Given the highly cognitive nature of this study, it is also notable that all the realted work is "previous benchmarks people have done" and none of it is the cognitive literature that produced the chosen assessments or justified why they measure the studied constructs.

I like the use of consistant and controlled opponants in the interactive scenarios and this facilitates a more robust and reliable comparison. However, its unclear to what extent devining the behaviour / mental state of these agents taxes ToM - for example for the rule based ones it is a matter of establishing which rule dictates their answer. It is important to cross reference how these tasks are being used and intepretted with the congnitive science literature.

Most importantly, It is not clear for all the evaluatons what the LLM is expected to do, and how this is scored. There needs to be much much more detail on teh tasks themselves and the methods generally. Without knowing what these different tasks require, how do we interpret that the same models pass some with flying colours while catastrophically failing others? There is also no review of these inconsistencies and what they might mean in the dicussion.

minor comments:

Why is "socialisation" always italicised?
Table 3 - I think the blue and red are the wrong way around and also most columns dont have a blue highlight.

**Strengths:**

I think its very possible that the authors have done some very sound experimental work here. however its difficult to assess whether or not this is the case based on what has been written (see weaknesses).

I like the use of consistant and controlled opponants in the interactive scenarios and this facilitates a more robust and reliable comparison.

I like the new world and counterfactual tasks and I think the results on these have the potential to be a good contribution.

**Weaknesses:**

Most importantly, It is not clear for all the evaluatons what the LLM is expected to do, and how this is scored. For example, while there is much description fo teh number guessing game, the actual ToM TASK is not described - what information is the LLM supposed to be providing and based on what? Similar for blackjack - "For the Blackjack scenario, we conducted a manual evaluation." - of what? There needs to be much much more detail on teh tasks themselves and the methods generally. Without knowing what these different tasks require, how do we interpret that the same models pass some with flying colours while catastrophically failing others? There is also no review of these inconsistencies and what they might mean in the dicussion - beyond who is better at what task.  The critiques as to experimental details are somewhat alleviated by the contents of teh appendix (although these are never referenced in the text) however only partially. First, only a subset of tasks are decribed, second the ones described still do not provide enought information: For e.g.  giving the rules of the game blackjack does not explain why this is considered to be a ToM task. Much of this information should be in the main text, and the "related work" section must include literature on the use of these tasks as ToM tests (including why they are considered to assess ToM). Other than the tasks that were adapted from existing ToM third-person vigniettes, it is not clear how the specific task instances were generated (for the static tasks) and what (if any) measures were taken to ensure robustness (e.g. systematic variation)

The fact that one version of each model was run once means that it is impossible to tell whether any chance / difference in performance is robust. Even when temperature is set at zero there are minor variations.

**Questions:**

The authors need to acknowledge that what they are investigating is a cognitive process, they are not simply providing a benchmark for how impressive a system is, but are claiming to measure the cognitive process of theory of mind - it therefore matters what tasks they are using and why they are considered tests of this capabiltiy. Further, its important to interpret, rather than simply present the results.

**Details Of Ethics Concerns:**

This paper provides a human baseline with little to know information about the humans and how they were assessed.

---

### Note · Authors · 2024-12-16

**Comment:**

Thanks for your suggestions！

**Withdrawal Confirmation:**

I have read and agree with the venue's withdrawal policy on behalf of myself and my co-authors.